# The Association of Circulating L-Carnitine, γ-Butyrobetaine and Trimethylamine N-Oxide Levels with Gastric Cancer

**DOI:** 10.3390/diagnostics13071341

**Published:** 2023-04-04

**Authors:** Ilmārs Stonāns, Jelizaveta Kuzmina, Inese Poļaka, Solveiga Grīnberga, Eduards Sevostjanovs, Edgars Liepiņš, Ilona Aleksandraviča, Daiga Šantare, Arnis Kiršners, Roberts Škapars, Andrejs Pčolkins, Ivars Tolmanis, Armands Sīviņš, Mārcis Leja, Maija Dambrova

**Affiliations:** 1Institute of Clinical and Preventive Medicine, University of Latvia, LV-1079 Riga, Latvia; 2Mass Spectrometry Group, Latvian Institute of Organic Synthesis, LV-1006 Riga, Latvia; 3Laboratory of Pharmaceutical Pharmacology, Latvian Institute of Organic Synthesis, LV-1006 Riga, Latvia; 4Riga East University Hospital, LV-1038 Riga, Latvia; 5Faculty of Medicine, University of Latvia, LV-1004 Riga, Latvia; 6Digestive Diseases Centre GASTRO, LV-1586 Riga, Latvia

**Keywords:** gastric cancer, L-carnitine, γ-butyrobetaine, trimethylamine N-oxide, diagnostic, biomarker, metabolite

## Abstract

Our study aimed to evaluate the association between gastric cancer (GC) and higher concentrations of the metabolites L-carnitine, γ-butyrobetaine (GBB) and gut microbiota-mediated trimethylamine N-oxide (TMAO) in the circulation. There is evidence suggesting that higher levels of TMAO and its precursors in blood can be indicative of either a higher risk of malignancy or indeed its presence; however, GC has not been studied in this regard until now. Our study included 83 controls without high-risk stomach lesions and 105 GC cases. Blood serum L-carnitine, GBB and TMAO levels were measured by ultra-high-performance liquid chromatography–mass spectrometry (UPLC/MS/MS). Although there were no significant differences between female control and GC groups, we found a significant difference in circulating levels of metabolites between the male control group and the male GC group, with median levels of L-carnitine reaching 30.22 (25.78–37.57) nmol/mL vs. 37.38 (32.73–42.61) nmol/mL (*p* < 0.001), GBB–0.79 (0.73–0.97) nmol/mL vs. 0.97 (0.78–1.16) nmol/mL (*p* < 0.05) and TMAO–2.49 (2.00–2.97) nmol/mL vs. 3.12 (2.08–5.83) nmol/mL (*p* < 0.05). Thus, our study demonstrated the association between higher blood levels of L-carnitine, GBB, TMAO and GC in males, but not in females. Furthermore, correlations of any two investigated metabolites were stronger in the GC groups of both genders in comparison to the control groups. Our findings reveal the potential role of L-carnitine, GBB and TMAO in GC and suggest metabolic differences between genders. In addition, the logistic regression analysis revealed that the only significant factor in terms of predicting whether the patient belonged to the control or to the GC group was the blood level of L-carnitine in males only. Hence, carnitine might be important as a biomarker or a risk factor for GC, especially in males.

## 1. Introduction

According to the global cancer statistics in 2020, gastric cancer (GC) remains one of the five leading malignancies in terms of incidence [1]. Over a million new cases are diagnosed yearly around the world, with men being at nearly twice as at risk from the disease as women [2]. A large proportion of GC cases are still diagnosed at advanced stages when the prognosis of the disease is pessimistic. The implementation of population-based prophylactic measures and screening has shown some promise [2,3]. Prophylactic measures to reduce GC burden are increasingly utilized and include dietary changes with regard to the consumption of salty and processed foods and lifestyle changes with avoidance of obesity and smoking and alcohol intake as well as *Helicobacter pylori* (*H. pylori*) eradication therapy [2,4,5]. Invasive GC screening techniques include an endoscopic evaluation. Non-invasive and complimentary screening consists of *H. pylori* testing and the evaluation of biomarkers such as gastrin-17 and pepsinogen. In Europe, especially in high-incidence areas (e.g., Eastern Europe), a population-based *H. pylori* test-and-treat strategy is underway in terms of the Accelerating Gastric Cancer Reduction in Europe through the *H. pylori* Eradication (EUROHELICAN) project [4]. Although prophylactic and screening measures are being implemented, the incidence of GC is still predicted to increase by 62% until 2040 [2,3]. Further scope to prevent or diagnose the disease early remains important. Some progress in understanding the link between gut microbiota, metabolic processes and cancer has been made in recent decades, opening a new opportunity for finding prognostic or therapeutic targets.

Gut microbiota is the collection of microorganisms that colonize the gastrointestinal tract making it highly diverse. The homeostasis of the gut microbiota is believed to be one of the important mediators of human health [6]. As far as in the early 20th Century, Hewetson et al. described the role of the intestinal microbiome in the activation of inflammation [7]. Since then, much has been done to better understand this interaction. There is a growing body of evidence about the influence of gut microbiota on the pathogenesis of various chronic diseases, namely atherosclerosis, diabetes, chronic kidney disease and cancer, among others [6,8]. For instance, Liu et al. have confirmed that colon cancer tissue specimens have different prevailing microbial colonies from the healthy surrounding tissue samples [9]. The heterogeneity of microbiota within colonic adenocarcinoma or precancerous adenoma showed a significant correlation with the malignization potential of precancerous lesions. Furthermore, this finding was consistent in subjects from different geographic areas of the world [9]. The microbiome of the colon is thought to play an important role in the origin of digestive cancers [10]. Specific microbiome and molecular changes of colonic microflora have been identified in individuals with cancer. Even hereditary cancer syndromes (e.g., Lynch syndrome) show an association with the gut microbiota [9,10].

Gastric microbiota was also reported to be altered within the gastric precancerous lesion development cascade [11,12]. There are findings denoting that the cancerogenic potential of well-known gastric bacteria, *H. pylori*, can be enhanced by the gastric microbiome and vice versa [11]. Indeed, persistent infection has shown an association with reduced gastric microbial diversity, e.g., the depletion of *Acinotobacter*, *Firmicutes* and *Bacteroides* spp., leading to a reduced mucous protection, disrupted immune response and potential of cancerogenesis [13]. Additionally, due to the reduced acidity of the gastric environment, chronic *H. pylori* infection can indirectly lead to permission for more microorganisms to pass through the acid barrier and, therefore, colonise the distal colon [13]. Finally, antibacterial therapy seems to influence both gastric and colonic microbiota in the short and long term by interrupting its composition and, therefore, metabolic activity. Some studies concluded that *H. pylori* eradication results in a higher relative colonic abundance of *Proteobacteria* and decreased diversity of microbiome [14]. To summarise, there is evidence of a complex crosstalk between gastric and colonic microbiota, inflammation and cancerogenesis.

After finding mechanisms denoting the association between the gut microbiome and cancer, many researchers have focused on identifying specific microbiota-dependant metabolites that could potentially be related to cancerogenesis. One such metabolite is trimethylamine N-oxide (TMAO). TMAO is synthesized in multiple steps. Two main sources of TMAO are L-carnitine and choline [15,16,17]. After ingestion, both metabolites are processed into trimethylamine (TMA) by gut microbiota. Choline is converted directly into TMA, whereas L-carnitine either undergoes an intermediate stage of transformation into γ-butyrobetaine (GBB) or can be directly transformed into TMA by gut microflora as well [17]. The next step is the absorption and transformation of TMA by the liver enzyme system into the TMAO [17,18].

TMAO is considered a cardiometabolic risk factor. In recent decades, a growing body of evidence has emerged showing that elevated levels of TMAO in the blood contribute to the development of atherosclerosis and cardiovascular events [19,20,21]. More recent studies established a certain link between TMAO and cancer [22]. Individuals with higher TMAO levels in systemic circulation demonstrated a higher risk of developing colorectal cancer [22,23]. Additionally, patients with colon cancer demonstrated higher TMAO levels than healthy controls [22,23,24]. These findings suggest an influence of gut microbiota composition and its metabolic activity on cancerogenesis. Furthermore, Liu et al. reported the results of a case-control cross-sectional study where an increased risk of liver cancer was registered in study participants with higher concentrations of TMAO in blood specimens [8]. In a case-control study with 130 pancreatic cancer patients, Hang et al. reported a link between elevated TMAO levels and pancreatic cancer in red meat consumers compared to vegetarians [25].

There are no data available about the possible association between levels of L-carnitine, GBB and TMAO in the bloodstream and GC. To investigate this problem, in our study we hypothesized that elevated levels of all three metabolites can be observed in GC cases, compared to individuals without high-risk stomach lesions.

## 2. Materials and Methods

### 2.1. Overall Design

All cases included in the study cohort were selected from the biobank sample collection managed in collaboration between the Institute of Clinical and Preventive Medicine of the University of Latvia and the Oncology Centre of Latvia, Riga East Clinical University Hospital. Subjects with a confirmed diagnosis of gastric adenocarcinoma at clinical T3 and T4 stages were included [26]. The controls were also selected from the biobank and all of them had undergone upper endoscopy at the Digestive Diseases Centre GASTRO to assure the absence of high-risk stomach lesions. The control group had normal gastric mucosa or insignificant atrophic changes according to the Operative Link of Gastritis Assessment (OLGA 0–1) [27]. Concentrations of L-carnitine, GBB and TMAO in the blood specimens were measured and compared between the control group and GC group in overall, male and female study populations.

### 2.2. Study Population

Biobanked blood samples of the study subjects were obtained before the surgical, radiological or any other GC intervention. Overall, 209 samples were analyzed encompassing 93 controls and 116 GC cases. Blood samples containing meldonium (24) were excluded. Meldonium is a cardiometabolic drug that lowers TMAO concentration through increased urinary excretion [28]. To account for the meldonium-induced effects on the blood TMAO concentration, samples from patients taking this medication were excluded from the data analysis.

### 2.3. Measurement of Levels of L-Carnitine, GBB and TMAO by UPLC/MS/MS

The concentrations of L-carnitine, GBB, TMAO and meldonium in human serum samples were measured using the UPLC/MS/MS method, as previously described [29,30] with minor modifications. Sample preparation consisted of simple protein precipitation with acetonitrile–methanol solution. As an internal standard, we used 3-(2,2-dimethyl-2-prop-1-yl-hydrazinium)propionate for all calculations. Briefly, 480 mL of an acetonitrile-methanol mixture (3:1, *v*/*v*) containing internal standard was added to 20 mL of serum sample. Samples were centrifuged at 11,000× *g* for 10 min to precipitate proteins. The cleared supernatants were removed and diluted (1:9, *v*/*v*) with the acetonitrile–methanol mixture (3:1, *v*/*v*) and injected into the UPLC/MS/MS system (Shimadzu LCMS-8060NX, Shimadzu, Japan). Chromatographic separation was performed on a BEH HILIC (1.7 µm, 2.1 × 100 mm) column (Waters Corp., Wilmslow, UK) at a flow rate of 0.25 mL/min. The composition of the mobile phase, namely acetonitrile with 10 mm aqueous ammonium acetate (pH 4), varied linearly from 75% to 55% of acetonitrile. TMAO, carnitine, GBB and meldonium were quantified by monitoring the specific transitions for each compound. Applied analytical procedures provided fair separation of all the analytes of interest in one run.

### 2.4. Statistical Analysis 

Descriptive statistics of the study cohort and biomarker values (differences and correlations) were analyzed with SPSS version 22.0. Data distribution was non-parametric, therefore the median, first quartile and third quartile (Mdn (Q1–Q3)) values were used as measures of variability. Differences in biomarker values and patient characteristics (age) in GC and control groups and in gender groups were calculated using the Mann–Whitney U test and the correlation of biomarkers and patient characteristics was assessed by Pearson’s correlation (R). Logistic regression for male and female groups was created to evaluate if the metabolites were significant factors in predicting whether the participant belonged to the control or GC group (R^2^). The difference between the groups was considered significant if the *p*-value was <0.05 (2-tailed).

## 3. Results

Overall, 93 controls and 116 GC cases were initially collected for the analysis. Furthermore, 24 samples were excluded from the data analysis because of meldonium presence. Finally, 83 controls and 105 GC cases remained in the study group for further analysis. The distribution of gender and age in the groups is summarized in Table 1. There was a higher prevalence of males in the GC group, whereas females dominated in the control group. The distribution of age was consistent between the control and GC groups and between the genders. In the control group, mild gastric atrophy (OLGA 1) was recorded in more females than males, showing 88.5% and 64.5%, respectively. Stage T4 was reported in 71.1% of females and 65.0% of males (Table 1).

Median concentrations of L-carnitine, GBB and TMAO in blood samples of controls and GC cases were compared in the combined gender group (Figure 1, upper row). Levels of L-carnitine were lower in the control group than in GC cases: 31.53 (26.87–37.77) nmol/mL and 35.69 (31.38–41.08) nmol/mL, respectively (*p* < 0.001) (Figure 1). Concentrations of GBB were recorded at 0.73 (0.66–0.82) nmol/mL in the control group and 0.84 (0.69–1.09) nmol/mL in the GC group (*p* < 0.001). Levels of TMAO did not show a significant difference between the two groups, with 2.65 (2.00–3.66) and 3.02 (2.06–4.99) nmol/mL for the control group and the GC group, respectively (*p* = 0.064) (Figure 1).

In females (Figure 1, lower row), median L-carnitine levels were 33.08 (27.30–37.86) nmol/mL in the control group and 35.07 (30.57–39.64) nmol/mL in the GC group (*p* = 0.259). Median GBB levels were 0.70 (0.62–0.76) nmol/mL in the control group and 0.76 (0.62–0.88) nmol/mL in the GC group (*p* = 0.154). Median TMAO levels were 2.66 (1.95–4.06) nmol/mL in the control group and 3.02 (1.86–4.38) nmol/mL for GC cases (*p* = 0.675). Thus, no significant differences were recorded between the control and GC groups in females.

In males (Figure 1, lower row), the median L-carnitine concentration was reported as 30.22 (25.78–37.53) nmol/mL in the control group and as 37.38 (32.73–42.61) nmol/mL in the GC group (*p* < 0.001). Median GBB levels were 0.79 (0.73–0.97) nmol/mL in the control group and 0.97 (0.78–1.16) nmol/mL in the GC group (*p* = 0.008). Median TMAO levels of 2.49 (2.00–2.97) nmol/mL in the control group and 3.12 (2.08–5.83) nmol/mL in the GC group were also significantly different (*p* = 0.036). Hence, the levels of all measured markers (L-carnitine, GBB and TMAO) were significantly higher in males with GC than in controls.

Pearson correlation coefficient values of L-carnitine, GBB and TMAO concentrations in the control group and GC cases were calculated (Table 2). In the control group, the only weak positive correlation was registered in females between L-carnitine and GBB levels (R = 0.29, *p* < 0.05). No significant correlation was found between remaining metabolite levels in the control group (Table 2). 

A moderate positive correlation between L-carnitine and GBB levels was recorded in both females (R = 0.53, *p* < 0.001) and males (R = 0.47, *p* < 0.001). A weak positive correlation was recorded between GBB and TMAO in females (R = 0.34, *p* < 0.05) and males (R = 0.27, *p* < 0.05) and between L-carnitine and TMAO in females (R = 0.36, *p* < 0.05) and males (R = 0.28, *p* < 0.05). Overall, every two of all three metabolites showed significant positive correlations in the GC group in both genders (Table 2). 

The logistic regression model for men showed that L-carnitine (R^2^ = 26.5%) was a significant factor (*p* < 0.001) in terms of predicting whether the patient belongs to the control or to the GC group, whereas GBB and TMAO were not significant (*p* = 0.583 and *p* = 0.223, respectively). At the same time, a logistic regression model based on L-carnitine, GBB and TMAO concentrations in the female group to predict the control or GC group did not show statistical significance for any metabolite (*p* = 0.077, *p* = 0.432 and *p* = 0.701, respectively).

## 4. Discussion 

In our study, we found significantly higher concentrations of L-carnitine, GBB and TMAO in the blood samples of men with GC when compared to healthy controls. Females did not demonstrate a significant difference in metabolite levels between the same groups. Gender difference in the consumption and metabolism of certain metabolites has been discussed in the literature. Overall, men are considered to have higher L-carnitine levels in the bloodstream than women [31]. One of the reasons is that on average, men have a higher consumption of foods rich in TMAO precursor carnitine, namely meat, dairy products and certain types of fish [32]. After ingestion, L-carnitine undergoes a transformation into GBB and TMA by gut microorganisms. Therefore, the consumption of more L-carnitine-enriched foods can result in higher levels of related metabolites (GBB and TMAO) in the bloodstream [19]. Another reason for the difference in L-carnitine concentrations among genders is their absorption variability. A study by Liepinsh et al. demonstrated 10% lower L-carnitine concentrations in females than males regardless of the consumption of foods rich in carnitine (e.g., red meat) [29]. 

Overall, a higher dietary intake of red and processed meat has been linked to a higher risk of cancer development, including GC [33]. Therefore, both products are included in the list of carcinogens by the World Health Organisation (WHO), classifying red meat as Group 2A and processed meat as Group 1 carcinogens [33,34]. While we had no data on the dietary patterns of our study population, L-carnitine is known to be mainly obtained from meat products, thus omnivores were reported to have significantly higher L-carnitine levels in the circulation than vegetarians in many studies [35]. However, there is contrasting research showing that circulating L-carnitine levels in vegetarians can be the same as in omnivores or even higher, due to the endogenous L-carnitine synthesis and carnitine obtained from plant-based foods or biological supplements [36]. In addition, there are studies confirming that omnivores have a higher capacity of producing TMAO from its precursors than vegetarians, mainly due to the differences identified in the gut microbiota composition [35,37].

Higher L-carnitine, GBB and TMAO concentrations in males with GC compared to healthy controls were found in our study population (Figure 1). For women, we demonstrated a tendency towards the same rise but could not prove its significance (Figure 1). Moreover, L-carnitine was a significant factor for predicting whether the male study subject belongs to the control or GC group, according to the logistic regression model. Various mechanisms can potentially explain raised concentrations of metabolites in GC cases. One such mechanism was described in two separate studies by Console et al. and Melone et al. In both studies, the authors describe the dependence of cancer cells on L-carnitine as one of the main energy resources via fatty acid metabolism. There is evidence of the activation and increased presence of transporters regulating carnitine traffic in cancer cells and in the plasma [38,39]. The overexpression of carnitine transporters in tumor tissues was associated with cancer cell growth, progression and development into more aggressive tumor types [40]. This process of activation of the carnitine pathway was called “cancer metabolic plasticity”, and was supported by some other authors who suggested it as a possible target for cancer therapies in the future [41,42]. Another study of 991 matched case-control pairs aimed to evaluate the role of carnitine in breast cancer development [43]. Increased circulating butyrylcarnitine levels in this study were associated with increased breast cancer risk, although malonylcarnitine, decenoylcarnitine and decadienolcarnitine showed protective effects against breast malignancy [43]. Some studies have concluded that malnutrition associated with cancer or anti-cancer treatment might decrease carnitine levels [39,44,45,46]. Overall, if an increased demand of carnitine in cancer cells for energy production is ensured by high L-carnitine concentrations in blood, more rapid GC growth could be expected. Therefore, if GC groups are characterized by higher levels of carnitine [42] they would be at a significantly higher cancer risk. However, it is important to note that these studies are limited in their scope and do not provide conclusive evidence that carnitine directly causes GC. More research is needed to better understand the relationship between carnitine and GC.

Dietary patterns with a higher intake of TMA- and TMAO-containing food (e.g., sea fish) have been reported to elevate TMAO concentrations in the bloodstream [22]. While some studies in mice models demonstrate a TMAO immunostimulatory effect, improved response to immune checkpoint inhibitors [47] and some protective effects on cellular proteins under stress conditions [48], TMAO is mainly associated with elevated cardiometabolic risks. In addition, higher levels of TMAO in systemic circulation have been linked to the risk of cancer development. A large study conducted by Bae et al. amongst 835 matched case-control female pairs found a three times higher colorectal cancer risk in women with increased blood TMAO concentrations [22]. Other authors looked into the risk of liver cancer and TMAO, where an increase in TMAO levels showed a positive association with the disease [8]. In addition, some scholars have confirmed a link between higher TMAO levels and the risk of colorectal and prostate cancers [49]. Liu et al. demonstrated the role of preoperative TMAO increases as prognostic tools for colorectal cancer [24]. Furthermore, there are reports of a positive association between TMAO levels and prostate cancer [33]. A genetic link between higher TMAO production and colorectal cancer was described by Xu et al. [50]. Genes encoding liver enzymes (flavin-containing monooxygenase, FMO) that oxidize TMA into TMAO were recorded among nearly ten other gene alterations, thereby linking increased TMAO production with colorectal cancer risk [50].

Apart from higher L-carnitine, GBB and TMAO levels in the male GC group, we have been able to register stronger positive correlations between any two of three metabolites in the GC group for both genders. While we have not evaluated the gut microflora of our study group, there is evidence arising from the literature that cancer, especially gastrointestinal malignancies, is associated with alterations in colonic microbiota, leading to the higher production of TMA and its further oxidation into TMAO [51]. In one of the recent studies, males with higher TMAO levels in blood demonstrated lower gut microbial diversity and a higher abundance of *Firmicutes* in their mucosa of large intestines [52,53]. This finding was confirmed in another study conducted by Clara E et al. where a higher colonic abundance of *Firmicutes* and *Bacteroidetes* in men was associated with an increase in blood TMAO levels after ingestion of its precursors, choline or carnitine, demonstrating the role of microbiota in determining how L-carnitine is further processed by microorganisms [54]. More studies confirmed the association of microbiota with TMAO production resulting in higher colorectal, breast, and gastric cancer risks [55]. Gut microbiota composition is a very sensitive entity that can be influenced by various factors that include gender, metabolic state and comorbidities. For instance, males and females within different BMI groups demonstrate different proportions of *Firmicutes* and *Bacteroides* (F/B ratio) in their gastrointestinal tract. Although the link is not yet very well established, it is apparent that the disequilibration of gut microbiota can result in increased TMA production and, sequentially, its oxidation into TMAO [56].

Apart from bacterial strains residing in the gastrointestinal tract, a gut–blood barrier was described as a variable affecting the absorption of various metabolites, including L-carnitine, GBB and TMA. Tools to assess the gut–blood barrier permeability are now investigated and put into practice [57,58]. There are data from animal models indicating that heart failure, for instance, results in increased gut–blood barrier permeability and the higher absorption of metabolites, including TMAO precursors [59].

## 5. Conclusions

Our findings demonstrate the potential role of L-carnitine, GBB and TMAO in GC and suggest metabolic differences between genders. In addition, the logistic regression analysis revealed that the only significant factor in terms of predicting whether the patient belonged to the control group or to the GC group was the blood level of L-carnitine in males only. Hence, carnitine might be important as a biomarker or as a risk factor for GC, especially in males.

## Figures and Tables

**Figure 1 diagnostics-13-01341-f001:**
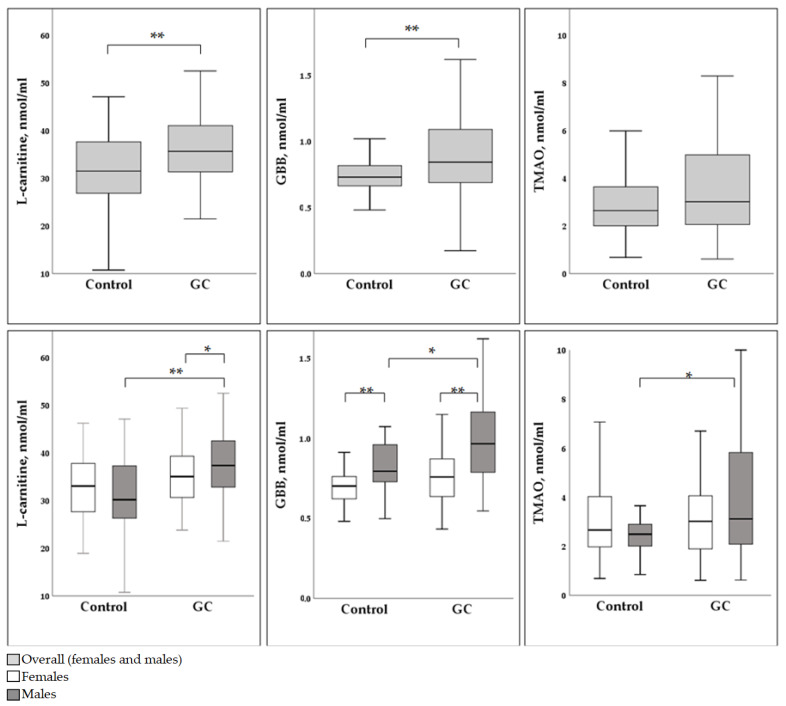
Serum concentrations of L-carnitine, GBB and TMAO in the control group and the gastric cancer group. (**Upper row**)—median concentrations in an overall group. (**Lower row**)—median concentrations in female and male groups. GBB—γ-butyrobetaine; TMAO—trimethylamine N-oxide levels; GC—gastric cnacer; * *p* < 0.05; ** *p* < 0.001.

**Table 1 diagnostics-13-01341-t001:** Descriptive statistics of the study population.

	Control (*N* = 83)		GC Cases (*N* = 105)	
	Females	Males	Females	Males
Number, *N* (%)	52 (62.6)	31 (37.4)	45 (42.9)	60 (57.1)
Age, mean ± SD, years	66.83 ± 9.89	61.23 ± 13.81	64.13 ± 11.12	64.3 ± 10.30
BMI, mean ± SD, kg/m^2^	30.37 ± 5.35	26.11 ± 4.94	26.09 ± 5.50	27.02 ± 4.66
T stage of GC				
T3, *N* (%)	N/A	N/A	13 (28.9)	21 (35.0)
T4, *N* (%)	N/A	N/A	32 (71.1)	39 (65.0)
Grade of gastric atrophy				
OLGA 0 (no atrophy), *N* (%)	6 (11.5)	11 (35.5)	N/A	N/A
OLGA 1 (mild atrophy), *N* (%)	46 (88.5)	20 (64.5)	N/A	N/A

GC—Gastric cancer; SD—standard deviation.; BMI—Body Mass Index; OLGA—Operative Link for Gastritis Assessment.

**Table 2 diagnostics-13-01341-t002:** Pearson correlation coefficients illustrating correlations between biomarker values L-carnitine, GBB and TMAO in the control group and gastric cancer cases.

Females							
Controls					GC		
R	L-carnitine	GBB	TMAO	R	L-carnitine	GBB	TMAO
L-carnitine	1	0.29 *	−0.03	L-carnitine	1	0.53 **	0.36 *
GBB	0.29 *	1	0.13	GBB	0.53 **	1	0.34 *
TMAO	−0.03	0.13	1	TMAO	0.36 *	0.34 *	1
Males							
Controls					GC		
R	L-carnitine	GBB	TMAO	R	L-carnitine	GBB	TMAO
L-carnitine	1	0.05	−0.03	L-carnitine	1	0.47 **	0.28 *****
GBB	0.05	1	0.20	GBB	0.47 **	1	0.27 *****
TMAO	−0.03	0.20	1	TMAO	0.28 *****	0.27 *	1
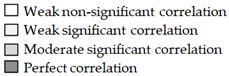							

GC—gastric cancer; R—Pearson correlation coefficient; GBB—γ-butyrobetaine; TMAO—trimethylamine N-oxide levels; * *p* < 0.05; ** *p* < 0.001.

## Data Availability

The data presented in this study are available on request from the corresponding author I.S. The data are not publicly available due to information that could compromise the privacy of research participants.

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
