# Peer review of "The Association of Circulating L-Carnitine, γ-Butyrobetaine and Trimethylamine N-Oxide Levels with Gastric Cancer"

_diagnostics, 2023, doi:10.3390/diagnostics13071341_

Round 1
Reviewer 1 Report
Although several reviews about the diagnostics of gastric cancer have been already published, the discussion on the markers of gastric carcinoma in this paper seems to be original. However I have the following suggestions/comments and hope the authors can address them in the review.
Minor revision
1. Some authors showed that the ADH/ALDH activities are higher in tumor cells than in normal gstric tissue, suggesting that isoenzymes of ADH may play an important role in carcinogenesis. Among all tested classes of ADH isoenzymes, only class IV had higher activity in the serum of patients with gastric cancer:
a) Jelski Wojciech, Chrostek Lech, Szmitkowski Maciej: The activity of class I, III and IV of alcohol dehydrogenase isoenzymes and aldehyde dehydrogenase in gastric cancer. Dig. Dis. Sci 2007, 52, 531-535.
b) Jelski Wojciech, Chrostek Lech, Zalewski Bogdan, Szmitkowski Maciej: Alcohol dehydrogenase (ADH) isoenzymes and aldehyde dehydrogenase (ALDH) activity in the sera of patients with gastric cancer. Dig. Dis. Sci 2008, 53, 2101-2105
c) Jelski Wojciech, Orywal Karolina, Łaniewska Magdalena, Szmitkowski Maciej: The diagnostic value of alcohol dehydrogenase (ADH) isoenzymes and aldehyde dehydrogenase (ALDH) measurement in the sera of gastric cancer patients. Clin.Exp. Med. 2010, 4, 215-219
Please discuss (5-6 sentences)
Reviewer 2 Report
1. Dietary patterns between vegetarian and omnivore should be considered as one influence factor.
2. Gastric microbiota has been reported to be involved in the cascade of gastric precancerous lesion development, particularly H. pylori. H. pylori should be considered as one influence factor.
3. What is the relationship between the concentration of metabolites (L-carnitine, GBB and TMAO) and cancer stages (I~IV)?
4. Authors should provide the No of IRB license.
Reviewer 3 Report
The manuscript is interesting for the readers and its topic suitable for the diagnostics. Nevertheless, some point can be taken. Authors should be also discussing limitations and potential applicability of their study. Also, role carnitine in the tumour biology should be more mentioned. For the inspiration I can recommend below works.
Han Y, Yoo HJ, Jee SH, Lee JH. High serum levels of L-carnitine and citric acid negatively correlated with alkaline phosphatase are detectable in Koreans before gastric cancer onset. Metabolomics. 2022 Jul 28;18(8):62. doi: 10.1007/s11306-022-01922-7.
GV, Jordão AA Jr, dos Santos JS, Marchini JS. Lower carnitine plasma values from malnutrition cancer patients. J Gastrointest Cancer. 2013 Sep;44(3):362-5. doi: 10.1007/s12029-013-9497-3. PMID: 23609166.
Takagi A, Hawke P, Tokuda S, Toda T, Higashizono K, Nagai E, Watanabe M, Nakatani E, Kanemoto H, Oba N. Serum carnitine as a biomarker of sarcopenia and nutritional status in preoperative gastrointestinal cancer patients. J Cachexia Sarcopenia Muscle. 2022 Feb;13(1):287-295. doi: 10.1002/jcsm.12906. Epub 2021 Dec 22. Erratum in: J Cachexia Sarcopenia Muscle. 2023 Jan 25;: PMID: 34939358; PMCID: PMC8818668.
Morgell A, Reisz JA, Ateeb Z, Davanian H, Reinsbach SE, Halimi A, Gaiser R, Valente R, Arnelo U, Del Chiaro M, Chen MS, D'Alessandro A. Metabolic Characterization of Plasma and Cyst Fluid from Cystic Precursors to Pancreatic Cancer Patients Reveal Metabolic Signatures of Bacterial Infection. J Proteome Res. 2021 May 7;20(5):2725-2738. doi: 10.1021/acs.jproteome.1c00018. Epub 2021 Mar 15. PMID: 33720736.
Chen T, Wu G, Hu H, Wu C. Enhanced fatty acid oxidation mediated by CPT1C promotes gastric cancer progression. J Gastrointest Oncol. 2020 Aug;11(4):695-707. doi: 10.21037/jgo-20-157. PMID: 32953153; PMCID: PMC7475321.
Kawai A, Matsumoto H, Endou Y, Honda Y, Kubota H, Higashida M, Hirai T. Repeated Combined Chemotherapy with Cisplatin Lowers Carnitine Levels in Gastric Cancer Patients. Ann Nutr Metab. 2017;71(3-4):261-265. doi: 10.1159/000485808. Epub 2017 Dec 13. PMID: 29237151.
Wang X, Wang J, Wang Z, Wang Q, Li H. Dynamic monitoring of plasma amino acids and carnitine during chemotherapy of patients with alimentary canal malignancies and its clinical value. Onco Targets Ther. 2015 Aug 7;8:1989-96. doi: 10.2147/OTT.S86562. PMID: 26300648; PMCID: PMC4535544.
Round 2
Reviewer 2 Report
Authors have reposed to reviewers and revised this manuscript well based on suggestions and comments of reviewers.